# Determinants of left ventricular structure, filling and long axis function in systemic sclerosis

**Roger E. Peverill** [1]*, **Gene-Siew Ngian**[2,3], **Catherine Mylrea**[1], **Joanne Sahhar**[2,3]

**1** Department of Medicine (School of Clinical Sciences at Monash Health), Monash Cardiovascular Research Centre, Monash Heart, Monash University and Monash Health, Clayton, Victoria, Australia, **2** Rheumatology Department, Monash Health, Monash University, Clayton, Victoria, Australia, **3** Department of Medicine (School of Clinical Sciences at Monash Health), Monash University, Clayton, Victoria, Australia

* roger.peverill@monash.edu

## Abstract

### Background

Abnormalities of left ventricular (LV) structure, filling and long-axis function have all been reported in subjects with systemic sclerosis (SSc) and a normal LV ejection fraction (EF), but previous study findings have not been consistent. The aim of this study was to identify factors which could have confounded the analyses in previous studies of SSc, and in particular to consider the variables of body surface area (BSA), sex, age, heart rate, blood pressure (BP), disease duration (DD), disease type (limited versus diffuse) and interstitial lung disease (ILD).

### Methods

Echocardiography was performed on 100 subjects with SSc (79 women; age 56±15 years) with a LVEF $\geq$50% and free of pulmonary arterial hypertension, coronary artery disease, more than mild valvular heart disease and atrial fibrillation. Measurements were performed of the LV end-diastolic dimension (LVEDD) and septal wall thickness (SWT), the transmitral Doppler E, A and deceleration time (DT), and the peak systolic (s') and early diastolic (e') LV long-axis velocities. Multivariate analyses were performed to investigate correlations of the above LV variables with BSA, sex, age, heart rate, BP, DD, disease type, and the presence of ILD.

### Results

DD varied between 0.1 and 41.2 years, 25% had diffuse and 75% had limited disease, and 37% had ILD. SWT and LVEDD were positively correlated with BSA, SWT was also positively correlated with age and larger in males, and LVEDD was larger in diffuse disease. Age was positively correlated with A and DT, and inversely correlated with E and E/A, and heart rate was inversely correlated with E and E/A. None of E, A, E/A, or DT were independently associated with DD or disease type. Septal and lateral LV wall s' and e' were all inversely correlated with age, and there was a small independent contribution to the prediction of

**Competing interests:** The authors have declared that no competing interests exist.

**Abbreviations:** A, transmitral atrial flow peak velocity; BP, blood pressure; BMI, body mass index; BSA, body surface area; DD, disease duration; DT, deceleration time; E—, transmitral early diastolic peak flow velocity; e', peak velocity of mitral annular early diastolic motion; ILD, interstitial lung disease; LV, left ventricular; LVEDD, left ventricular end-diastolic diameter; LVEF, left ventricular ejection fraction; LVM, left ventricular mass; PAH, pulmonary arterial hypertension; PWT, posterior wall thickness; RWT, relative wall thickness; SSc, systemic sclerosis; SWT, septal wall thickness; s', peak velocity of mitral annular systolic motion; TDI, tissue Doppler imaging.

lateral s' from DD, but no association of either s' or e' with disease type. The presence of ILD was not a predictor of any of the LV variables.

## Conclusion

In SSc there are associations of sex, body size, age and disease type with LV structural variables, of age and heart rate with E/A, and of age with both systolic and early diastolic LV long-axis velocities. Appropriate adjustment for these variables could help to resolve current uncertainties regarding SSc effects on the left ventricle.

## Introduction

Systemic sclerosis (SSc) is a chronic, multi-system disease characterized by fibrosis and vascular damage [1], for which clinical involvement of the heart is associated with a worse prognosis [2]. Left ventricular (LV) ejection fraction (EF) is generally normal in SSc [3–5], but there have been reports that SSc can affect LV structure and function in the presence of a normal LVEF. Reported LV abnormalities include increases in septal wall thickness (SWT), relative wall thickness (RWT) and LV mass (LVM) index [5–7], decreases in the transmitral E [8] and E/A ratio [8–13], and increases in A [7, 8, 10, 13–15], and the deceleration time (DT) [8, 12, 16], and reductions in peak pulse wave tissue Doppler imaging (TDI) velocities of long-axis motion in both systole (s') [15, 17] and early diastole (e') [11, 17–20]. However, there have also been a number of negative studies with regard to all of the above LV abnormalities [6–24], and there have been few attempts to provide explanations for the divergences in findings.

The aim of this study was to identify possible determinants of LV structure, transmitral flow and LV long-axis function which could have confounded previous studies in SSc, but may not have been taken into account. Body surface area (BSA), sex, age, heart rate and blood pressure (BP) have all been considered based on data from healthy populations indicating associations with LV echocardiographic variables. For example, in healthy subjects increasing age is associated with increases in wall thickness and RWT [25, 26] and accompanied by lower transmitral E, E/A, s' and e', a higher A, and a longer DT [27]. Sex and body size also have recognized effects on LV structure, filling and long-axis velocities in healthy subjects [26, 28–36], and transmitral Doppler and LV long-axis peak velocities have associations with heart rate [28, 35, 37], and BP [28, 34, 36, 38, 39]. The possibility of LV effects of the SSc-specific features of interstitial lung disease (ILD), disease type (limited versus diffuse) and disease duration (DD) has also been investigated.

## Methods

### Subjects

Subjects were identified from a dedicated SSc clinic at a tertiary hospital (Monash Medical Centre, Monash Health) between February 2010 and February 2014. Subjects were over 18 years of age and satisfied the ACR/EULAR 2013 Classification Criteria for diagnosis of SSc [40]. The LeRoy and Medsger criteria were used for classification into limited and diffuse types [41]. At this clinic the subjects undergo annual screening for cardiac disease and pulmonary arterial hypertension (PAH) with a transthoracic echocardiogram, and for ILD with respiratory function testing and selective performance of high resolution computed tomography of the chest. ILD has been defined for this study as the presence of interstitial

abnormalities on high-resolution computed tomography of the chest. Patients suspected of having PAH (estimated systolic pulmonary artery pressure [PAP] $\geq$ 40 mmHg at echocardiography, diffusing capacity of the lung for carbon monoxide [DLCO] $\leq$ 50% with forced vital capacity > 85%, or with DLCO $\geq$ 20% and/or unexplained dyspnoea) are considered for right heart catheterisation (RHC) [42]. PAH was defined as a mean PAP $\geq$25 mmHg in association with a pulmonary arterial wedge pressure <15 mmHg and subjects found to have PAH were not included in this study. Exclusion criteria for the current study were clinical evidence of coronary artery disease (based on past history, symptoms, or electrocardiographic or echocardiographic changes suggesting ischemia or myocardial infarction), a LVEF < 50% [43], reduction of right ventricular ejection fraction, more than mild valvular stenosis or regurgitation, atrial fibrillation or flutter, diabetes, stroke, symptomatic peripheral vascular disease or an estimated glomerular filtration rate (eGFR) <30 mL/min/1.73 $m^2$. Patients were defined as having chronic kidney disease (CKD) if they had an eGFR of 30–60 mL/min/1.73 $m^2$. The study was approved by the Monash Health Human Research and Ethics Committee, the need for written consent was waived, and analysis was performed on anonymized data.

## Echocardiography

Two-dimensional echocardiography was performed by an experienced ultrasonographer using a Vivid 7 ultrasound machine (Vingmed, General Electric, Milwaukee, WI, USA) and studies were stored digitally for measurement off-line using Xcelera V1.2 L4 SP2 (Philips, Amsterdam, The Netherlands). Brachial blood pressure (BP) was measured at the time of echocardiogram. M-mode recordings of the left ventricle were made from the parasternal long-axis view according to the American Society of Echocardiography guidelines [44]. Standard LV M-mode measurements were made of LV end-diastolic diameter (LVEDD), SWT and posterior wall thickness (PWT). Relative wall thickness was calculated as 2 x PWT/ LVEDD. LV mass has not been reported in this study due to concern about the effects of older age on the accuracy of the M-mode formula, and also based on the recognized discrepancy between echocardiographic and cardiac magnetic resonance findings [33, 45, 46]. 2D loops from the 4-chamber, 2-chamber and apical long-axis views were recorded for assessment of LV regional contraction and for measurement of LVEF using the biplane method of discs. 4- and 2-chamber loops were recorded after optimisation on the atria and used for measurement of left atrial (LA) area and length, with calculation of LA volume performed by the biplane area-length method [47]. LA volume was indexed to BSA (LA volume index; LAVI), and categorized as increased if >34 mL/$m^2$.

LV inflow velocities were recorded using pulsed-wave Doppler in the apical 4-chamber view with the sample volume at the level of the mitral leaflet tips, and the peak transmitral E and A wave velocities were measured off-line. If there was onset of the atrial flow signal during the deceleration phase of the early diastolic flow signal (i.e. partial fusion of the signals) then the A has been calculated as the peak velocity of the atrial signal minus the velocity at the time of the onset of the atrial signal. Pulsed wave TDI was recorded at the mitral annular level during end-expiration apnoea with particular attention made to orientation of the cursor parallel to the long-axes of the septal and lateral walls. Off-line measurements of 3 consecutive cardiac cycles of s' and e' were performed as previously described [34, 36]. Continuous wave Doppler of the tricuspid regurgitation (TR) jet was performed in all available imaging planes to enable identification of the peak velocity. Systolic PA pressure was calculated as estimated RV pressure (4 x the square of the peak velocity of the TR jet) + the right atrial pressure (estimated from the diameter of the inferior vena cava and its variation with respiration) [48]. Diastolic function was assessed as recommended by the American Society of Echocardiography (ASE)/

European Association of Cardiovascular Imaging (EACVI) in their 2016 guidelines using the following variables: septal and lateral e', E/e' (average e' from septal and lateral walls), LAVI, and the TR peak velocity [27]. Abnormal findings were a septal e' < 7 cm/s or lateral e' < 10 cm/s, an E/e' > 14, a LAVI > 34 mL/m2, and a TR velocity > 2.8 m/s.

## Statistics

Statistical analysis was performed using Systat V13 (Systat Software, Chicago, IL, USA). Continuous variables are expressed as mean ± standard deviation (SD) or median [range] and compared with an unpaired t test, or a non-parametric test if the variable was not normally distributed. Independent variables which were not normally distributed were log transformed before inclusion in regression analyses. For univariate linear regression analysis the correlation coefficient (r) is provided and for multivariate analyses, the partial standardized correlation coefficient (β) is provided. The coefficient of determination has been adjusted for the number of terms in multivariate models (adjusted $r^2$) and used to estimate the percentage of the variance of a dependent variable explained by that model. The decisions regarding testing of independent variables in multivariate models were based on the aims of the study, variables were added step-wise, and variables which did not contribute to an increase in the adjusted $r^2$ for a model were removed. Dummy variables were used for sex (female = 1, male = 0), type of disease (diffuse = 1, limited = 0) and ILD (present = 1, absent = 0). Multiple logistic regression was performed for selected bivariate LV variables and the area under the curve (AUC) for the receiver operating characteristic curve is provided. A p value of <0.05 was considered significant.

## Results

The relevant characteristics of the study group are shown in Table 1. The range of ages in the cohort was 22–83 years and the range of DD was 0.1–41.2 years, 79% were females, 25% had diffuse and 75% had limited disease, 37% had ILD and 31% had a history of hypertension. The group with diffuse disease were younger than those with limited disease (48±16 v 59±13 years, p = 0.003), and also had a younger age at onset of symptoms (40±14 v 50±14 years, p = 0.007),

**Table 1. Clinical characteristics of subjects with systemic sclerosis (n = 100).**

| | |
|---|---|
| Female | 79 (79%) |
| Age (years) | 56.2±14.6 |
| Hypertension | 31 (31%) |
| Systolic BP at time of study (mmHg) | 131±19 |
| Diastolic BP at time of study (mmHg) | 74±10 |
| Heart rate at time of study (beats/min) | 67±8 |
| Body surface area ($m^2$) | 1.70±0.17 |
| Body mass index (kg/$m^2$) | 25.6±5.1 |
| Chronic kidney disease (eGFR 30–60 mL/min/1.73 $m^2$) | 4 (4%) |
| Age at onset of symptoms (years) | 47.4±14.5 |
| Disease duration (years) | 6.0 [0.1–41.2] |
| Diffuse disease | 25 (25%) |
| Anti-centromere Ab positive | 42 (42%) |
| Anti-Scl-70 Ab positive | 22 (22%) |
| Interstitial lung disease | 37 (37%) |

Results are shown as mean±SD or median [range].

but there was no significant difference in DD between these groups (7.1 [0.4–28.5] v 5.6 [0.1–41.0] years, p = 0.82). There were no differences in age, age at onset of symptoms or DD between those with and without ILD (p = 0.35, p = 0.59 and p = 0.46, respectively). Given that DD was not normally distributed, it was natural log transformed prior to its inclusion as an independent variable in regression models. The possibility of confounding in multivariate regression analyses due to colinearity between age, DD and BP was explored. Age was positively correlated with systolic BP (r = 0.56, p<0.001) and diastolic BP (r = 0.23, p = 0.024), log DD was positively correlated with age (r = 0.25, p = 0.011), but log DD was not correlated with either systolic or diastolic BP (p>0.10 for both).

## Left ventricular structural variables

The echocardiographic findings of the study group are shown in Table 2. Measurements of SWT and PWT, and calculation of RWT, were feasible in 98 subjects. A SWT or PWT >1.1 cm was only present in 3 subjects. In univariate linear regression analysis, LVEDD, SWT and PWT were all positively correlated with BSA, and smaller in females than males (p<0.05 for all), whereas RWT was not correlated with BSA or different based on sex (p>0.2 for both). Multivariate models of LV structural variables with age and log DD, after adjustment for sex and BSA (except for RWT), are shown in Table 3. Age was positively correlated with SWT, PWT and RWT, but not with LVEDD. Log DD was only positively correlated with RWT.

Multivariate modeling of LV structural variables was performed with the sequential addition of sex, BSA, age, log DD, disease type, systolic BP, ILD and history of hypertension, and the models of LVEDD and SWT with independent variables with a p<0.10 are shown in Table 4. Female sex was associated with a smaller SWT and PWT, BSA was positively correlated with LVEDD, SWT and PWT, and age was positively correlated with SWT, PWT and RWT. There was a borderline significant inverse correlation of log DD with LVEDD, but not with RWT, and the presence of diffuse disease was associated with a larger LVEDD. After adjustments for the above variables, there were no contributions to any of the models from systolic BP, a history of hypertension or ILD.

**Table 2. Echocardiographic variables in subjects with systemic sclerosis.**

| | |
|---|---|
| LVEDD (cm) | 4.7±0.4 |
| SWT (cm) | 0.9±0.1 |
| PWT (cm) | 0.8±0.1 |
| RWT | 0.35±0.05 |
| LVEF (%) | 60±5 |
| Left atrial volume index ($cm^2/m^2$) | 35±10 |
| E (cm/s) | 73±16 |
| A (cm/s) | 66±20 |
| E/A | 1.22±0.49 |
| DT (ms) | 194 [28–462] |
| Septal s' (cm/s) | 6.6±1.3 |
| Septal e' (cm/s) | 8.0±2.5 |
| Lateral s' (cm/s) | 8.1±1.9 |
| Lateral e' (cm/s) | 10.5±3.7 |
| E/Average e' | 8.5±3.0 |

Results are shown as mean±SD or median [range].

**Table 3. Correlations of age and disease duration with left ventricular structural, transmitral Doppler and left ventricular long-axis function variables in subjects with systemic sclerosis.**

|  | Age | | Log DD | |
|---|---|---|---|---|
|  | r/β | P | r/β | p |
| LVEDD* | -0.16 | 0.08 | -0.14 | 0.11 |
| SWT* | 0.45 | <0.001 | 0.11 | 0.28 |
| PWT* | 0.45 | 0.004 | 0.17 | 0.08 |
| RWT | 0.47 | <0.001 | 0.23 | 0.021 |
| E | -0.27 | 0.008 | -0.04 | 0.72 |
| A | 0.68 | <0.001 | 0.15 | 0.15 |
| E/A | -0.74 | <0.001 | -0.16 | 0.12 |
| Log DT | 0.38 | <0.001 | 0.19 | 0.057 |
| Septal s' | -0.52 | <0.001 | -0.27 | 0.006 |
| Septal e' | -0.78 | <0.001 | -0.17 | 0.09 |
| Lateral s' | -0.45 | <0.001 | -0.30 | 0.003 |
| Lateral e' | -0.81 | <0.001 | -0.23 | 0.022 |
| E/e' | 0.55 | <0.001 | 0.14 | 0.17 |

*With adjustment for sex and BSA.

## Correlates of Doppler and TDI variables

The univariate correlates of transmitral Doppler and TDI variables with age and log DD are shown in Table 3. DT was not normally distributed and was natural log transformed prior to the analysis. There were positive correlations of age with A and log DT, inverse correlations of age with E and E/A, and there was a weak positive correlation of log DD with A. Age was inversely correlated with septal and lateral s' and e' and positively correlated with E/e'. Age alone accounted for more than 50% of the variances of septal and lateral e'. Log DD was also inversely correlated with septal s' and lateral s' and e', but not with septal e'. Univariate correlations of transmitral Doppler and TDI variables with heart rate and systolic BP were also investigated. Heart rate was an inverse correlate of E (r = -0.24, p = 0.017), but was not correlated with any of the other transmitral or TDI variables. Systolic BP was a positive correlate of A (r = 0.53, p<0.001) and E/e' (r = 0.47, p<0.001) and an inverse correlate of E/A (r = -0.43, p<0.001), septal s' (r = -0.42, p<0.001), septal e' (r = -0.48, p<0.001), lateral s' (r = -0.42, p<0.001) and lateral e' (r = -0.55, p<0.001).

Multivariate modeling of transmitral Doppler and TDI variables was performed with the sequential addition of age, log DD, sex, heart rate, systolic BP, disease type, ILD, and hypertension history, and models of independent variables with more than one dependent variable with a p<0.10 are shown in Table 5. Age was a correlate of all the transmitral Doppler and

**Table 4. Multivariate models of selected left ventricular structural variables in subjects with systemic sclerosis.**

| Dependent variable | Independent variable | Univariate r | β in multivariate model | p value in multivariate model | Cumulative Adjusted r² |
|---|---|---|---|---|---|
| **LVEDD** | BSA | 0.46 | 0.42 | <0.001 | 0.20 |
|  | Log DD | -0.22 | -0.17 | 0.055 | 0.22 |
|  | Diffuse disease | 0.30 | 0.27 | 0.002 | 0.29 |
| **SWT** | Female sex | -0.28 | -0.25 | 0.006 | 0.07 |
|  | BSA | 0.26 | 0.25 | 0.007 | 0.10 |
|  | Age | 0.40 | 0.45 | <0.001 | 0.29 |

**Table 5. Selected Multivariate models of selected LV filling and LV long-axis TDI variables.**

| Dependent variable | Independent variable | Univariate r | β in multivariate model | p value in multivariate model | Cumulative Adjusted $r^2$ |
|---|---|---|---|---|---|
| E | Age | -0.27 | -0.30 | 0.003 | 0.06 |
| | Heart rate | -0.24 | -0.27 | 0.005 | 0.13 |
| A | Age | 0.68 | 0.55 | <0.001 | 0.46 |
| | Systolic BP | 0.53 | 0.23 | 0.013 | 0.48 |
| E/A | Age | -0.74 | -0.73 | <0.001 | 0.55 |
| | Heart rate | -0.12 | -0.23 | 0.001 | 0.59 |
| | Diffuse disease | 0.31 | 0.14 | 0.05 | 0.60 |
| Lateral s' | Age | -0.45 | -0.28 | 0.001 | 0.20 |
| | Systolic BP | -0.42 | -0.30 | 0.005 | 0.29 |
| | Log DD | -0.45 | -0.18 | 0.045 | 0.31 |
| E/e' | Age | 0.55 | 0.41 | <0.001 | 0.29 |
| | Systolic BP | 0.32 | 0.25 | 0.017 | 0.33 |

TDI variables, sex did not contribute to any of the models, and log DD was an inverse correlate of lateral s'. Heart rate was a correlate of E and E/A, and systolic BP was a correlate of A, lateral s' and E/e'. A diagnosis of diffuse disease made a borderline significant contribution to the model of E/A. There were no independent contributions to any of the models from a diagnosis of ILD or hypertension.

### Logistic regression analysis of abnormal diastolic variables

E/A was <1.0 in 23% of subjects, there was a low septal or lateral e' in 54%, E/e' was >14 in 8%, the left atrium was dilated in 48% and the TR peak velocity was >2.8 m/s in 3%. Using the 2016 ASE/EACVI criteria, LV diastolic dysfunction was present in 8%, indeterminate diastolic dysfunction in 27% and elevation of mean LA pressure in 6%. Subjects with diastolic dysfunction were compared to the group with normal diastolic function. The diastolic dysfunction group was older at the time of the study (74±5 v 50±14 years, p<0.001) and at onset of symptoms (59±15 v 42±13 years, p = 0.017), and had a higher DD (10.3 [6.4–36.8] v 5.2 [0.2–33.2], p = 0.02). On multivariate logistic regression analysis excluding those subjects with indeterminate diastolic function, and with independent variables of age, log DD, systolic BP, disease type, ILD and history of hypertension, older age was the only significant predictor of the presence of diastolic dysfunction (p = 0.002, AUC = 0.95). Including the same independent variables in multivariate logistic regression analyses of the total subject group, older age was also the only significant predictor of a low E/A (p<0.001, AUC = 0.84), whereas older age (p<0.001) and higher systolic BP (p = 0.004) were both predictors of a low e' (AUC = 0.90).

### Discussion

This study identified a number of variables with the potential to have confounded the findings of previous studies in which LV structural and filling variables and long-axis velocities have been compared in SSc and control subjects. In multivariate analyses, BSA, sex and age were each independent correlates of at least one of the LV structural variables, age was an independent correlate of all the Doppler and TDI variables, and heart rate was an independent correlate of some of the transmitral Doppler variables. These findings are not unexpected as they have also been reported in healthy populations, but they are important because they could not be assumed to exist in SSc, they have not always been reported in studies comparing SSc and control subjects (See Tables 6 and 7 for a selection of relevant studies), and even when

**Table 6. A summary of the subject characteristics and available demographic data in a selection of systemic sclerosis echocardiographic studies in which LV structural or filling variables or long-axis velocities have been compared to control subjects.**

| | SSc group (n) | SSc group (M/F) | SSc group age (years) | Limited/ Diffuse disease | Subjects with PAH | Control group (n) | Control group (M/F) | Control group age (years) | Heart rate | BSA or height & weight data | BP data | DD in years |
|---|---|---|---|---|---|---|---|---|---|---|---|---|
| Kazzam [6] | 30 | 15/15 | 55±2 [range 25–77] | N/A | N | 48 | 26/22 | 55±2 [range 25–77] | Y | Y | Y | 6 (0.5–23) |
| Henein [17] | 34 | 9/25 | 49±12 | N/A | N | 21 | 14/7 | 51±11 | N | N | N | N |
| Armstrong [16] | 35 | 2/33 | 48±15 | 21/14 | Y | 35 | N/A | 45±14 | Y | Y | N | 10 (0–27) |
| Aguglia [7] | 124 | 18/106 | 52±12 | 67/57 | Y | 41 | 4/37 | 53±14 | Y | N | Y | 11±8 |
| Maione [8] | 77 | 6/71 | 54±11 | 19/17* | N | 45 | 4/41 | 52±10 | Y | N | N | 18±9 |
| D'Andrea [21] | 23 | 3/20 | 56±8 | 12/11 | N | 25 | 5/20 | 55±9 | Y | Y | Y | N |
| D'Andrea [18] | 33 | 9/24 | 56±8 | 18/15 | N | 30 | 7/23 | 55±9 | Y | Y | Y | N |
| Meune [9] | 100 | 14/86 | 54±14 | 58/42 | N | 26 | 5/21 | 50±13 | N in controls | N | N | 8±8 |
| Mele [19] | 35 | 1/34 | 55±13 | 23/12 | N | 35 | 0/35 | 53±9 | N | N | N | 10±8 |
| Kepez [20] | 27 | 1/26 | 46±10 | 18/9 | N | 26 | 1/25 | 44±10 | Y | N | Y | 10 (3–18) $ |
| Poanta [10] | 20 | 2/18 | 52±10 | 5/15 | N | 15 | 3/12 | 48±13 | Y | N | N | 47±56! |
| Lee [14] | 35 | 9/26 | 49±13 | N/A | N | 35 | 10/25 | 52±8 | N | N | Y | 10±2! |
| Yiu [22] | 104 | 24/80 | 54±12 | 51/53 | Y | 37 | 7/27 | 54±10 | N | N | N | 6±4 |
| Plazak [11] | 46 | 3/43 | 55 [range 24–73] | 25/21 | N | 30 | 0/30 | 52 [range 38–57] | N | N | N | 15 (2–32) |
| Aktoz [12] | 26 | 2/24 | 49 [range 21–71] | N/A | N | 24 | 2/22 | 46 [range 22–56] | Y | N | Y | 60 [range 12–276]! |
| Faludi [15] | 34 | 3/31 | 57±12 | 27/7 | Y | 23 | 6/17 | 53±10 | N | Y | N | Mean ~ 12 |
| Ciurzynski [13] | 111 | 10/101 | 54±14 | 64/47 | N | 21 | 3/18 | 49±11 | Y | Y | Y | 9±12 (1–25) |
| Karadag [23] | 47 | 5/42 | 52±12 | 33/14 | N | 36 | 3/33 | 49±8 | N | Y | Y | 9±6 |
| Agoston [24] | 42 | 2/40 | 50±14 | 35/7 | N | 42 | 2/40 | 49±13 | N | N | N | N |

* 41 were classed as intermediate disease

$ median (25th-75th percentile)

!months; N/A—not available; N—information not provided; Y—information provided.

reported, they have not always been taken into account when comparing the results of SSc and control subjects.

Body size was a determinant of LV structural variables (LVEDD, SWT and PWT) in SSc subjects in the present study, a finding consistent with data from population studies in subjects free from known cardiovascular disease [26, 32]. Male sex was a determinant of larger SWT and PWT independent of BSA in SSc subjects, and this finding is also consistent with data from healthy subjects [26, 33]. While there is no specific reason why body size and sex should not also be determinants of LV structural variables in SSc, neither could it be assumed that there would be the same relationships in SSc as in healthy subjects. Furthermore, our findings highlight the importance in considering both body size and sex in studies of LV structure in SSc. However, only 7/19 of the studies shown in Table 6 provided any data on body size in SSc

**Table 7. A summary of the findings of a selection of systemic sclerosis echocardiographic studies in which LV structural or filling variables or long-axis velocities have been compared to control subjects.**

| | LVEF | LVEDD | Wall thickness | RWT | LVM index | E | A | E/A | DT | Septal s' | Septal e' | Lateral s' | Lateral e' | E/e' |
|---|---|---|---|---|---|---|---|---|---|---|---|---|---|---|
| Kazzam [6] | | N | ↑ | ↑ | ↑ | | | | | | | | | |
| Henein [17] | N | N | N | | | | | | | ↓ | ↓ | ↓ | ↓ | |
| Armstrong [16] | | N | N | N | N | | | N | ↑ | | | | | |
| Aguglia [7] | | N | ↑ | | ↑ | N | ↑ | N | N | | | | | |
| Maione [8] | | N | N | | | ↓ | ↑ | ↓ | ↑ | | | | | |
| D'Andrea [21] | | N | N | | N | N | N | N | N | | | N | N | |
| D'Andrea [18] | N | N | N | | N | N | N | N | N | N | ↓ | N | ↓ | |
| Meune [9] | N | N | N | | | | | ↓ | | N* | | N* | N | N |
| Mele [19] | N | N | N | | N | N | N | N | N | N* | ↓* | N* | ↓* | ↑ |
| Kepez [20] | N | N | | | | N | N | N | N | N | N | N | ↓ | |
| Poanta [10] | N | N | N | | | N | ↑ | ↓** | | | | N | N | ↑ ** |
| Lee [14] | N | N | N | | | N | ↑ | N | | N | N | N | | ↑ |
| Yiu [22] | N | | | | | | | | | | | N | ↓ | ↑ |
| Plazak [11] | N | | | | | | | ↓ | | N* | N* | N* | N* | |
| Aktoz [12] | N | | | | N | | | ↓ | ↑ | N* | N* | N* | N* | ↑ |
| Faludi [15] | N | N | | | ↑ | N | ↑ | N | | | | ↓ | ↓ | ↑ |
| Ciurzynski [13] | ↓ | | | | | N | ↑ | ↓ | N | | ↓ | | N | N |
| Karadag [23] | N | N | | | N | N | N | N | N | | | | N | N |
| Agoston [24] | N | | | | | | | | | | | | | ↑ |

*Average of septal and lateral TDI velocities presented

** $p = 0.05$.

and control subjects, and so differences in the body size of SSc and control subjects could not be evaluated. Moreover, sex matching of SSc and control subjects has rarely been exact in previous SSc studies, the ratio of males to females in the SSc cohort has varied between the studies, and in one study the sex of the control subjects was not even reported [16]. Importantly, even if there was exact sex matching and a similar mean BSA in SSc and control subjects in a study, this cannot exclude differences in the relationships of sex with BSA between the SSc and control groups which could confound the findings. However, knowing this possibility, the potential for such confounding can be addressed through performance of a multivariate linear regression analysis with adjustment for both BSA and sex when relevant [49].

Age was an independent predictor of SWT, PWT and RWT in the present study, findings which are consistent with previous reports in healthy subjects [25, 26, 33], and which are particularly relevant in the study of the left ventricle in SSc given that the ages of affected subjects can range from young adults to those >80 years. A lack of correlation of LVEDD with age in SSc subjects after adjustment for sex and BSA in the present study is also consistent with previous studies in healthy subjects where adjustment for BSA was performed [26, 50]. Although echocardiographic data in healthy subjects has suggested that LVM index increases with age [33], this finding has not been confirmed in cardiac magnetic resonance studies [45, 46]. This discrepancy may be because of an intrinsic limitation of the echocardiographic cubed M-mode formula [51], and may be of particular importance in older age groups in which there is a change in shape of the left ventricle [52]. In view of this, determinants of LVM were not considered in the present study and LVM findings in previous echocardiographic SSc studies will be of less certain significance than the measurements of LVEDD, SWT and RWT which do not rely on complex geometric assumptions. Although the mean ages of SSc and control

subjects were not significantly different in any of the SSc studies listed in Table 6, standard statistical testing does not test for differences in age range or SDs, and also cannot test for grouping of variables which might affect the findings (e.g. between age, DD, sex and BSA). Indeed, an important, albeit not unexpected grouping was found in the current study, this being that subjects with diffuse disease were younger than those with limited disease.

The effects of SSc on the heart can progress over time, and while this may well not be a linear progression, it could be considered essential to take the duration of disease symptoms into account when determining SSc effects on the heart. However, DD has not always been reported in SSc studies, and even when it has been reported, it has sometimes been reported as mean and SD despite the numbers presented demonstrating DD values which are unlikely to have been normally distributed. There have been studies in which the relations of DD with LV echocardiographic variables have been evaluated, but age and disease type have generally not been considered within the same analysis. The potential importance of considering all 3 variables in the same analysis was highlighted in the present study as those with diffuse disease were younger, and there was a correlation of age with log DD (transformed because it was not normally distributed). DD had no more than a minor effect on LV structural functional variables in the present study, with a trend to a smaller LVEDD, independent of sex and BSA. While an effect of DD on LVEDD has not been previously reported, we could not find any previous study which specifically investigated this relationship. Our finding of a larger LVEDD in diffuse compared to limited disease has also not been previously reported to the best of our knowledge, but required a multivariate analysis, which has also not previously been performed. An effect of diffuse versus limited disease on LV structure is feasible given that there are a number of other differences in the expression of disease phenotype between the disease groups. Nevertheless, our findings regarding diffuse disease will need confirmation, particularly as they are not consistent with previous univariate comparisons [5].

Abnormalities of LV filling variables have been reported in SSc, but there has been no consistency in the findings of the published studies (see Table 7). In studies in healthy adults, transmitral Doppler variables are well recognized to be affected by age [28, 31], and have also been reported to be affected by sex, BP and heart rate [28, 31]. That these factors also affect Doppler variables in SSc was demonstrated in the present study, as age was an independent correlate of all the transmitral variables, heart rate had an inverse association with E and E/A, and there was a positive correlation of systolic BP with A. Although adjustment for all these factors could therefore be considered essential during investigations of LV filling in SSc subjects, such adjustment has not generally been performed. Indeed, heart rate and blood pressure data have not been provided in a number of previous studies. In multivariate models in the present study, with appropriate adjustments for age, heart rate and systolic BP, there were no contributions from either DD or disease type to the prediction of LV filling variables. Furthermore, in logistic regression analysis of a low E/A, the only significant predictor was age. Therefore, there was no supporting evidence from the present study that SSc has any effect on LV filling.

There are gradual reductions in LV long-axis s' and e' during healthy aging [34, 36, 53, 54], with decreases in e' becoming evident as early as the 3rd decade. The reduction in e' with age can be substantial over the normal life span, with a decrease of >50% between an age <30 years and an age >70 years [54]. An aging effect on long-axis function therefore needs to be considered in any investigation of the effect of SSc on LV long-axis function, particularly given that SSc can be manifest in individuals over the whole span of adult ages. There have also been reports of correlations of height, BSA and heart rate with s' [35], and of BP and BMI with e' [34, 36]. In the present study of subjects with SSc ranging in age from 22–83 years, age was inversely correlated with septal and lateral s' and septal and lateral e', and age explained the

major portion of the variances in e' for the septal and lateral walls (53% and 62%, respectively). That there was also a small additional contribution to the prediction of lateral s' from DD after inclusion of age in the model, provides some support for the presence of a myocardial effect of SSc on long-axis function, implying an effect of both the presence and the duration of the disease. The variability in DD between studies (Table 6) could therefore be a possible contributor to previous inconsistencies in the literature regarding whether SSc results in LV long-axis dysfunction (see Table 7). However, the effect of DD on s' was not consistent for the septal and lateral LV walls given that there was no effect of DD on septal s'. There was also a weak inverse correlation of log DD with lateral e' in the present study, but this was no longer significant when log DD was combined in a model with age. Although this does not exclude an effect of DD on e', it is consistent with the univariate correlation of DD with lateral e' being due to collinearity of DD with age. Previous studies which have investigated the effect of DD on LV long-axis function had discrepant findings and did not do multivariate analyses with adjustment for age [9, 10].

We found no correlations of BSA or heart rate with s', or of BMI with e', in the present study, but systolic BP at the time of the study was a univariate inverse correlate of septal and lateral s' and of septal and lateral e'. On multivariate analyses with adjustment for age, systolic BP was no longer a predictor of septal or lateral e', or of septal s', but was an independent inverse correlate of lateral s' and a positive correlate of E/e'. These findings are consistent with a portion of the effect of older age on s' and e' being via its association with a higher BP. An association of BP at the time of the study with e' has been previously reported in cross-sectional studies [34, 36], acute studies in healthy individuals [55], and an association of BP with e' has also been reported in longer-term studies using BP lowering drugs [56, 57]. These findings highlight the importance of measuring BP at the time of the study, and taking BP into consideration when comparing LV function variables in SSc and control groups. There was no evidence of an effect of disease type, the presence of ILD, or a history of hypertension on any of the long-axis variables in the present study.

Using the 2016 ASE/EASCI criteria [27], we identified LV diastolic dysfunction in 8%, indeterminate diastolic dysfunction in 27% and elevation of LA pressure in 6%. Age was the only independent predictor of the presence of diastolic dysfunction, but older age and higher systolic BP were independent predictors of a low e'. There have been previous cohort studies in SSc which have evaluated diastolic function [5, 58–60], and when determinants of diastolic function have been evaluated in these studies, all have found age to be a predictor, but other predictors have also been identified [58–60]. However, there has been a wide variation in the reported frequency of diastolic dysfunction in SSc (17–44%) [5, 58–60], each study has used different diagnostic criteria, and all of the previous studies have had less strict exclusion criteria than the present study.

There were a number of limitations of the present study. Our cohort was only of moderate size and there were relatively few subjects who met the 2016 ASE/EACI criteria for diastolic dysfunction, with the result that our study had limited ability to identify determinants of diastolic dysfunction other than age. On the other hand, it is of interest that only a small proportion of our SSc cohort met the current criteria for a diagnosis of diastolic dysfunction. Moreover, the moderate group size was at least partially compensated for by having restricted inclusion criteria, as by excluding subjects with PAH, manifest coronary artery disease, atrial arrhythmias, and more than mild valvular disease, we were more able to focus on the role of a number of specific variables in the absence of these potential confounding factors. As with all studies in SSc, there was a predominance of females in this study, and the low numbers of males will have limited the ability to detect sex-related differences in LV structure and function. Although manifest coronary artery disease was an exclusion criteria, asymptomatic

coronary disease could not be excluded as testing was not performed in asymptomatic subjects. However, even if unrecognized asymptomatic coronary artery disease was present in a proportion of our cohort, given the absence of regional hypokinesis, and that this was a study of images obtained at rest, it is unlikely to have influenced the main study findings. In multivariate analyses in which DD was being considered, greater variability in DD might have improved the ability of the study to detect effects of DD which were independent of age. However, there may be intrinsic limitations to having a SSc cohort with greater DD variability and that fits our study group criteria given that complications become more likely over time. Lastly, a large control group with exact matching of SSc subjects for age, sex and body size would have contributed additional information to our understanding of SSc effects on the left ventricle, but was not available for this study, and indeed, has generally not been present in previous SSc studies. On the other hand, a control group was not necessary to address the main aim of this study, which was to identify variables which could have confounded the results of previous studies.

In this cross-sectional study of SSc, we investigated the effects of a number of disease-specific and non-disease-specific variables on echocardiographic variables of LV structure, filling and long-axis function. We demonstrated associations of sex, BSA, age and disease type with LV structural variables, associations of age, heart rate and BP with LV filling variables, and associations of age with both systolic and early diastolic LV long-axis velocities. Therefore, a number of variables have been identified which may have varied between the SSc groups in different studies and between the SSc and control groups within studies, and thus could have influenced the results of these studies. An important implication of our findings is that multivariate analyses with adjustment for these variables would provide a means for investigating and potentially resolving previous divergent findings regarding LV structure, filling and long-axis function in SSc. Such an approach should not only be considered in future studies of SSc, but there could be a role for its use in the reanalysis of data from previous studies.

## Supporting information

**S1 Data. Scleroderma LV function data.**
(XLSX)

## Author Contributions

**Conceptualization:** Roger E. Peverill, Gene-Siew Ngian, Joanne Sahhar.

**Data curation:** Roger E. Peverill, Gene-Siew Ngian, Catherine Mylrea.

**Formal analysis:** Roger E. Peverill, Gene-Siew Ngian.

**Investigation:** Roger E. Peverill, Gene-Siew Ngian, Catherine Mylrea.

**Methodology:** Roger E. Peverill, Gene-Siew Ngian.

**Project administration:** Roger E. Peverill, Gene-Siew Ngian.

**Resources:** Roger E. Peverill, Gene-Siew Ngian.

**Software:** Roger E. Peverill, Gene-Siew Ngian.

**Supervision:** Roger E. Peverill.

**Validation:** Roger E. Peverill, Gene-Siew Ngian.

**Writing – original draft:** Roger E. Peverill.

**Writing – review & editing:** Roger E. Peverill, Gene-Siew Ngian, Catherine Mylrea, Joanne Sahhar.

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
