## [Decision Letter · Decision Letter 0]

8 Apr 2021

PONE-D-21-07257

Determinants of left ventricular structure, filling and long axis function in systemic sclerosis

PLOS ONE

Dear Dr. Peverill,

Thank you for submitting your manuscript to PLOS ONE. After careful consideration, we feel that it has merit but does not fully meet PLOS ONE’s publication criteria as it currently stands. Therefore, we invite you to submit a revised version of the manuscript that addresses the points raised during the review process.

This article is of potential interests and may provide novel insights into characterization of myocardial involvement in patients with systemic sclerosis, but all reviewers pointed out a number of critical issues that require amendment and improvement. Especially, lack of correction for multiple comparisons, lack of age and sex-matched healthy control group to compare, and lack of data on global longitudinal strain are critical issues of this study. We are happy to re-evaluate this article if the authors believe that they are able to address those important issues adequately in the revised version.

We look forward to receiving your revised manuscript.

Kind regards,

Masataka Kuwana, MD, PhD

Academic Editor

PLOS ONE

Journal Requirements:

Reviewers' comments:

Reviewer's Responses to Questions

**Comments to the Author**

1. Is the manuscript technically sound, and do the data support the conclusions?

Reviewer #1: Yes

Reviewer #2: Partly

Reviewer #3: Yes

2. Has the statistical analysis been performed appropriately and rigorously? 

Reviewer #1: Yes

Reviewer #2: Yes

Reviewer #3: Yes

3. Have the authors made all data underlying the findings in their manuscript fully available?

Reviewer #1: Yes

Reviewer #2: Yes

Reviewer #3: Yes

4. Is the manuscript presented in an intelligible fashion and written in standard English?

Reviewer #1: Yes

Reviewer #2: Yes

Reviewer #3: Yes

5. Review Comments to the Author

Reviewer #1: I am happy to have an opportunity to review an interesting manuscript written by Dr. Peverill, et al. regarding left ventricular abnormalities in systemic sclerosis. In my opinion, there are several problems that the authors should consider to revise.

1. The aim of this study was to explain why the previous relevant studies concluded differently. The authors stated in the conclusion that multivariate analyses adjusting for the potential confounding variables could resolve the previous discrepant findings, but it sounds strange. Conclusion should be novelty. Multivariate analysis is not likely to be the novelty of this study.

2. Please consider to reorder the tables. Table 1 and Table 2 are literature reviews. Since this is an original article, these tables should be behind other tables and only stated in the discussion part.

3. The number of the enrolled patients is missing in Table 3.

4. The definition of the abbreviations is missing in Table 4.

Reviewer #2: The present paper aims to solve current inconsistencies in potential associations between LV systolic and diastolic echo parameters and systemic sclerosis. Unfortunately, it falls short on several important issues, which should be discussed:

- The authors tested a plethora of different associations, without any attempt for multiple comparison correction. Having such a high number of potential correlations, it is natural that some of them could reach statistical significance by chance. Therefore, the proposed associations (LVEDD and BSA, SWT and age, LVEDD and diffuse disease, age and A, DT, and E/A, HR and E/A), which the authors advocate as useful in solving the current "dilemma", could have been produced by chance. A multiple comparison correction should be implemented for the whole statistical analysis to strengthen the results, otherwise, the discussion is not supported by the data and only tries to make sense of the hundreds of statistical tests performed.

- Regarding the vast majority of statistical associations tested who did not reach statistical significance, the authors should take into consideration that the sample size is very low (n=100) and therefore not reaching the p-value can be inconclusive. Nonetheless, throughout the text and the discussion, all p values > 0.05 were considered as lack of association, whereas the low statistical power should explain many of the results.

- Why was global longitudinal strain not considered? It is nowadays the best parameter for detecting subclinical progression of systemic sclerosis-related cardiomyopathy and can be retrieved offline. Moreover, it could have made the paper less obsolete.

- There are many large studies that details echo parameters that were not included in Table 2 (Eur J Prev Cardiol 2018; 25: 1598-1606; Eur Heart J Cardiovasc Imag 2012; 13: 863–870; Cardiovasc Ultrasound 2014; 12: 13 and probably many others). Therefore, the table is not complete and, therefore, not informative.

- One of the potential limitation of the present paper is the inclusion of patients with a history of disease from 0 to 40 years. This is a very heterogeneous population, and the effect over time of systemic sclerosis-related cardiomyopathy could not be linear (Eur J Prev Cardiol 2020; 27: 1876–1886), therefore making the presented results more difficult to interpret.

Reviewer #3: Determinants of left ventricular structure, filling, and long-axis function in systemic

sclerosis

A cross-sectional study evaluating the possible effect of disease type, DD, the presence of ILD, age, body size, sex, BP, heart rate, and history of hypertension on the LV structure, transmitral flow, and LV long-axis function.

Comments:

The main weakness of the study is the lack of age and sex-matched healthy control group to compare and investigate the relationship between the above-mentioned variables and left LV structure, transmitral flow, and LV long-axis function. The authors used the cut-off values for the determinants defined by 2016 ASE/EACI criteria, however, it would be better to include a control group concurrently to compare the results outside the normal limits.

1. Introduction:

In the introduction, the targetted subject of the study is explained in a very scattered way. By keeping the tables (Table 1 and 2) for discussion and giving a summary of the past studies, you might write a more impressive and summary introduction. I would kindly recommend you give a clear and concise insight into the topic for the introduction.

Language editing is recommended for the Introduction.

2. Methods:

In the first paragraph of the section “Subjects", you state that you enrolled the patients fulfilling the three different classification criteria. You should better use only one classification criteria, preferably the 2013 ACR/EULAR classification criteria. And for classifying the patients into limited and diffuse subtypes, it is better to use the LeRoy 1988* criteria.

* LEROY EC, BLACK C, FLEISCHMAJER R, et al.: Scleroderma (systemic sclerosis): classification, subsets, and pathogenesis. J Rheumatol 1988;15(2):202–205

3. Results

In Table 6, there are the results of multivariate regression analysis of two dependent variables (LVEDD and SWT). However, in the text above the table, you report the results about PWT and RWT. I suppose you included the variables with meaningful results in the table. However, you should mention this in the text or otherwise show all the results in Table 6. I have the same comments about Table 7.

4. Discussion

You should better emphasize the main and most important results of the study in the first paragraph of the Discussion. Considering the large amount of data analyzed in this study, it seems difficult to shorten the discussion. However, I would kindly recommend you to review the manuscript in terms of a brief and strong discussion by avoiding the unnecessary repetition of the results,

6. PLOS authors have the option to publish the peer review history of their article (what does this mean?). If published, this will include your full peer review and any attached files.

Reviewer #1: No

Reviewer #2: No

Reviewer #3: No

---

## [Author Response · Author response to Decision Letter 0]

28 Jul 2021

Reviewer #1: I am happy to have an opportunity to review an interesting manuscript written by Dr. Peverill, et al. regarding left ventricular abnormalities in systemic sclerosis. In my opinion, there are several problems that the authors should consider to revise.

1. The aim of this study was to explain why the previous relevant studies concluded differently. The authors stated in the conclusion that multivariate analyses adjusting for the potential confounding variables could resolve the previous discrepant findings, but it sounds strange. Conclusion should be novelty. Multivariate analysis is not likely to be the novelty of this study.

#Authors response. The aim of our study was to identify variables which might have confounded the comparison of SSc and control subjects in previous studies in SSc, and thus provide possible explanations for previous divergent findings with regard to LV M-mode measures, LV filling variables, s` and e`. A single study was never going to be able to resolve all previous discrepant findings. Our main conclusion was that there are a number of possible factors which could have varied between the studies and between the SSc subjects and controls which could have influenced the results of these studies. The importance of multivariate analyses to identify such factors was shown in the current study and an important implication of this is that multivariate analysis could be used in reanalysis of previous studies to adjust for the possible contributions of confounding factors. The statement about the potential value of multivariate analysis to adjust for the identified variables in the abstract has been modified to improve clarity. The identification of the variables is the new finding of this study, a potential utility of these findings is adjustment for their potential contribution to differences between SSc and control subjects using multivariate analysis. 

2. Please consider to reorder the tables. Table 1 and Table 2 are literature reviews. Since this is an original article, these tables should be behind other tables and only stated in the discussion part.

#Authors response. The Tables have been moved.

3. The number of the enrolled patients is missing in Table 3.

#Authors response.

This Table has now been renumbered as Table 1. The number of subjects is now included in the title of the table.

4. The definition of the abbreviations is missing in Table 4.

#Authors response.

Table 4 is now renumbered as Table 2. All the abbreviations are included at the beginning of the manuscript and our understanding is that this is acceptable to the editorial policy of PLOS One. 

Reviewer #2: The present paper aims to solve current inconsistencies in potential associations between LV systolic and diastolic echo parameters and systemic sclerosis. Unfortunately, it falls short on several important issues, which should be discussed:

- The authors tested a plethora of different associations, without any attempt for multiple comparison correction. Having such a high number of potential correlations, it is natural that some of them could reach statistical significance by chance. Therefore, the proposed associations (LVEDD and BSA, SWT and age, LVEDD and diffuse disease, age and A, DT, and E/A, HR and E/A), which the authors advocate as useful in solving the current "dilemma", could have been produced by chance. A multiple comparison correction should be implemented for the whole statistical analysis to strengthen the results, otherwise, the discussion is not supported by the data and only tries to make sense of the hundreds of statistical tests performed.

#Authors response. We agree that testing for multiple associations has the potential for false positive findings, particularly if a scatter gun approach to variable selection is used. However, there are some important distinctions to be made about this study with respect to the suggestion that we should have made adjustments for multiple comparisons. First, the main aim of this study was to identify possible factors which could have confounded the analysis of previous studies in SSc. Adjusting for multiple comparisons has less relevance in this situation. In other words, there are reasons to slightly favour the possibility of false positives compared to the possibility of false negatives in this circumstance. Second, the variables tested were selected based on previous reports in healthy subjects, previous reports in SSc subjects, or on a limited amount of hypotheses arising from plausible associations with SSc-disease specific features. Therefore, examples such as positive correlations of BSA with LVEDD, age with SWT, age with A, E/A and DT, and heart rate with E/A and DT could not be assumed in SSc, but were expected based on extensive data in healthy subjects, and thus are not likely to be due to chance. 

- Regarding the vast majority of statistical associations tested who did not reach statistical significance, the authors should take into consideration that the sample size is very low (n=100) and therefore not reaching the p-value can be inconclusive. Nonetheless, throughout the text and the discussion, all p values > 0.05 were considered as lack of association, whereas the low statistical power should explain many of the results.

#Authors response. We agree that a p value >0.05 does not mean that there is no association, only that there is no evidence for an association. We make no claims that we have proved that there are variables with no connection to LV structural and functional variables in SSc. However, a p value >0.05 in a regression analysis of 100 subjects does suggest that any genuine association missed due to insufficient numbers is unlikely to be a close one. 

- Why was global longitudinal strain not considered? It is nowadays the best parameter for detecting subclinical progression of systemic sclerosis-related cardiomyopathy and can be retrieved offline. Moreover, it could have made the paper less obsolete.

#Authors response. We agree that GLS is of interest in SSc but certainly do not agree that GLS makes all other LV variables obsolete. For example, GLS is a marker of long-axis contraction and thus provides no direct information about LV short axis diameter, wall thickness or diastolic function. Furthermore, even if GLS is abnormal in SSc it would still be important to understand the associations of GLS, e.g. s` and GLS would be expected to be correlated. Most pertinent for the current study, however, GLS was not included because it is important, this study was already long, and a study of GLS would require a substantial increase in the methodology, results and discussion in an already long manuscript.

- There are many large studies that details echo parameters that were not included in Table 2 (Eur J Prev Cardiol 2018; 25: 1598-1606; Eur Heart J Cardiovasc Imag 2012; 13: 863–870; Cardiovasc Ultrasound 2014; 12: 13 and probably many others). Therefore, the table is not complete and, therefore, not informative.

#Authors response. We agree that the list of studies is not complete. However, this was not a review paper and the aim of the Tables were to show that there are a substantial number of discrepancies between previous reports in SSc, and also to highlight that potentially relevant patient demographic data are missing from a number of previous studies. It would certainly be important if there was a study investigating the effects on the left ventricle of SSc that we had not mentioned, which had provided all the relevant information, and also adjusted for all potential confounding variables, but we are unaware of such a study. The first mention of the relevant tables (now renumbered 6 and 7) is now in the first paragraph of the discussion and it now makes the point that the tables include a selection of relevant studies.

- One of the potential limitation of the present paper is the inclusion of patients with a history of disease from 0 to 40 years. This is a very heterogeneous population, and the effect over time of systemic sclerosis-related cardiomyopathy could not be linear (Eur J Prev Cardiol 2020; 27: 1876–1886), therefore making the presented results more difficult to interpret.

#Authors response. We agree that there is heterogeneity in the study group. This heterogeneity has both benefits and hazards. If the aim is to determine whether there are correlations with disease duration, then variation in disease duration is necessary. However, we also agree that the effect of SSc over time may well not be linear. Age was correlated with log disease duration in the current study so it is important to consider this relationship when evaluating the potential effects of disease duration. A limitation of some previous studies which have investigated the effects of disease duration is that they have not adjusted for age. We would also point out that the limitations of the current study regarding assessment of the effects of disease duration are already mentioned in the discussion.

Reviewer #3: Determinants of left ventricular structure, filling, and long-axis function in systemic

sclerosis

A cross-sectional study evaluating the possible effect of disease type, DD, the presence of ILD, age, body size, sex, BP, heart rate, and history of hypertension on the LV structure, transmitral flow, and LV long-axis function.

Comments:

The main weakness of the study is the lack of age and sex-matched healthy control group to compare and investigate the relationship between the above-mentioned variables and left LV structure, transmitral flow, and LV long-axis function. The authors used the cut-off values for the determinants defined by 2016 ASE/EACI criteria, however, it would be better to include a control group concurrently to compare the results outside the normal limits.

#Authors response

The aim of this study was to identify variables which could have confounded previous echocardiographic studies in SSc given the inconsistency of previous LV findings. Our aim was not to compare SSc and control subjects. A study in which there was another comparison of SSc and control subjects would certainly be of value, but would not necessarily resolve all these discrepancies, particularly in view of the findings of the present study with respect to the possible confounding factors. Furthermore, a manuscript which addressed the aim of our study and also included a control group would be of unmanageable size. On the other hand, the identification of possible confounding factors provides options for re-analysis of previous studies with adjustment for these factors and for planning future studies which include control groups.

1. Introduction:

In the introduction, the targetted subject of the study is explained in a very scattered way. By keeping the tables (Table 1 and 2) for discussion and giving a summary of the past studies, you might write a more impressive and summary introduction. I would kindly recommend you give a clear and concise insight into the topic for the introduction.

Language editing is recommended for the Introduction.

#Authors response. Thank you for this suggestion. We have made a number of changes in an attempt to make the introduction more concise and aid in its clarity. 

2. Methods:

In the first paragraph of the section “Subjects", you state that you enrolled the patients fulfilling the three different classification criteria. You should better use only one classification criteria, preferably the 2013 ACR/EULAR classification criteria. And for classifying the patients into limited and diffuse subtypes, it is better to use the LeRoy 1988* criteria.

* LEROY EC, BLACK C, FLEISCHMAJER R, et al.: Scleroderma (systemic sclerosis): classification, subsets, and pathogenesis. J Rheumatol 1988;15(2):202–205

#Authors response.

The references have been changed as suggested. 

3. Results

In Table 6, there are the results of multivariate regression analysis of two dependent variables (LVEDD and SWT). However, in the text above the table, you report the results about PWT and RWT. I suppose you included the variables with meaningful results in the table. However, you should mention this in the text or otherwise show all the results in Table 6. I have the same comments about Table 7.

#Authors response.

These comments now refer to the renumbered Table 4 and Table 5. We have slightly changed the titles of these 2 tables to reflect that they are selected models. There were already statements in the text regarding the choice of the dependent variables. PWT was not included separately because the findings were similar to SWT. RWT was not included in the table because it was only correlated with age. Only independent variables with more than one dependent correlate were included in Table 5. The statement in the text has been rewritten to improve its clarity. 

4. Discussion

You should better emphasize the main and most important results of the study in the first paragraph of the Discussion. Considering the large amount of data analyzed in this study, it seems difficult to shorten the discussion. However, I would kindly recommend you to review the manuscript in terms of a brief and strong discussion by avoiding the unnecessary repetition of the results,

#Authors response. Thank you for these suggestions. A number of changes to the discussion have been made.

---

## [Decision Letter · Decision Letter 1]

17 Aug 2021

PONE-D-21-07257R1

Determinants of left ventricular structure, filling and long axis function in systemic sclerosis

PLOS ONE

Dear Dr. Peverill,

Thank you for submitting your manuscript to PLOS ONE. After careful consideration, we feel that it has merit but does not fully meet PLOS ONE’s publication criteria as it currently stands. Therefore, we invite you to submit a revised version of the manuscript that addresses the points raised during the review process.

The manuscript has been improved by revision, but one of the reviewers still has concern listed below:

It should be noted that in a disease such as systemic sclerosis, where clinical manifestations and disease activation are not always correlated with disease duration, confounding factors cannot be compared as easily as in the normal population. The authors should then have considered including some disease-specific variables (such as the modified Rodnan skin score or EUSTAR disease activity index) in their multivariate analysis. Since systemic sclerosis is a rare and special disease, making inferences as in studies in the normal population may result from not knowing the disease well enough. By agreing with the second reviewer, I think that some relationships can come out by chance without doing multiple comparison analysis. In my view, the authors' effort to show that the results of previous SSc studies were inadequate or inaccurate is more prominent than the purpose of reporting the results of their own study in an objective way. This turns into an effort to make the readers accept the accuracy of their own results rather than putting the originality of the article forward.

I ask the authors to improve the manuscript by trying to solve the issues as possible. 

We look forward to receiving your revised manuscript.

Kind regards,

Masataka Kuwana, MD, PhD

Academic Editor

PLOS ONE

Journal Requirements:

Reviewers' comments:

Reviewer's Responses to Questions

**Comments to the Author**

1. If the authors have adequately addressed your comments raised in a previous round of review and you feel that this manuscript is now acceptable for publication, you may indicate that here to bypass the “Comments to the Author” section, enter your conflict of interest statement in the “Confidential to Editor” section, and submit your "Accept" recommendation.

Reviewer #1: All comments have been addressed

Reviewer #3: (No Response)

2. Is the manuscript technically sound, and do the data support the conclusions?

Reviewer #1: Yes

Reviewer #3: No

3. Has the statistical analysis been performed appropriately and rigorously? 

Reviewer #1: Yes

Reviewer #3: No

4. Have the authors made all data underlying the findings in their manuscript fully available?

Reviewer #1: Yes

Reviewer #3: Yes

5. Is the manuscript presented in an intelligible fashion and written in standard English?

Reviewer #1: Yes

Reviewer #3: No

6. Review Comments to the Author

Reviewer #1: The authors successfully answered to my comments. I have no further comments. Thank you again for giving me an opportunity to review the paper.

Reviewer #3: (No Response)

7. PLOS authors have the option to publish the peer review history of their article (what does this mean?). If published, this will include your full peer review and any attached files.

Reviewer #1: **Yes: **Masaru Kato

Reviewer #3: No

---

## [Author Response · Author response to Decision Letter 1]

29 Sep 2021

Responses to Reviewer 3 comments

# It should be noted that in a disease such as systemic sclerosis, where clinical manifestations and disease activation are not always correlated with disease duration, confounding factors cannot be compared as easily as in the normal population. The authors should then have considered including some disease-specific variables (such as the modified Rodnan skin score or EUSTAR disease activity index) in their multivariate analysis. 

# Author’s response. We agree that clinical manifestations and disease activation may not be correlated with disease duration in systemic sclerosis (SSc), and also that there could well be factors which affect the left ventricle which are different to the normal population. The study does already include SSc disease-specific variables in the analysis other than disease duration, specifically disease type and interstitial lung disease. Furthermore, we already discuss the limitations of disease duration as an independent variable in the manuscript. Nevertheless, disease duration is a variable which has been reported to be a determinant of LV variables in previous studies and we believe that further testing of its association with LV variables in SSc was justified. Our study is certainly not the last word on the left ventricle in SSc, and disease-specific variables other than the ones we have studied and reported on could turn out to shed further light on the topic. However, the information requested regarding the Rodnan skin score and EUSTAR disease activity index is not available in all our subjects. Moreover, we fail to see how the absence of these variables invalidates either the originality, importance or the validity of our findings as currently presented.

# Since systemic sclerosis is a rare and special disease, making inferences as in studies in the normal population may result from not knowing the disease well enough. 

#Author’s response. We are not sure which inferences are being referred to by the reviewer, or which aspect of SSc the reviewer believes has not been understood by the authors. Moreover, we find this statement particularly surprising given that the main findings of the current study relate to a number of the same factors which affect left ventricular structural and functional variables in the normal population (e.g. age, body size, heart rate, blood pressure), and that we did find to have similar relationships with left ventricular variables in SSc. On the other hand, that we did not assume that this would be the case in SSc was part of the rationale for performing this study.

# By agreeing with the second reviewer, I think that some relationships can come out by chance without doing multiple comparison analysis. 

#Author’s response. We presume this statement relates to the issue of whether adjustments should have been made because multiple variables were tested in regression analyses. We did already address this question in a detailed response in our previous rebuttal but have restated and expanded this response below.

We agree that testing for multiple associations has the potential for false positive findings, particularly if a scatter gun approach to variable selection is used. However, the appropriateness of such adjustment depends on the aims and design of the particular study, and we believe that there are reasons why it is not necessary for the present study. First, the main aim of this study was to identify possible factors which could have confounded the analysis of previous studies in SSc, and we argue that adjusting for multiple comparisons has less relevance in this situation. In other words, there are reasons to favour the possibility of false positives compared to the possibility of false negatives in this circumstance. Second, there was no scatter gun approach to the choice of variables (i.e. a limited amount of independent variables were tested based on previous evidence). There are therefore also Bayesian issues given the considerable evidence for associations of the variables of age, sex, heart rate, blood pressure and body surface area with a number of left ventricular variables in the healthy population. In other words, our observations in SSc of correlations of BSA with LVEDD, age with SWT, age with A, E/A and DT, and heart rate with E/A and DT could not be assumed in SSc, but nevertheless were able to be postulated based on extensive data in healthy subjects, and thus are not likely to be due to chance. The SSc disease-specific variables were also carefully selected based on previous reports of their effects on left ventricular variables in SSc subjects (disease duration and interstitial lung disease) or because there was a plausible association with SSc disease-specific features (SSc type).

A possible explanation for any false positive finding in comparisons of 2 groups is that it has occurred due to unrecognized differences in one or more other variables between two groups which have had a direct effect on the dependent variable in question. It is thus worthwhile pointing out again that identifying such possible variables was the main aim of the present study. We also wish to emphasise that the positive associations reported in the current study were based on multivariable analyses which therefore provided adjustments for other contributing variables, and that the findings of such analyses are therefore more robust than positive associations that might have been present only in single variable analyses (see Slinker et al Circulation 2008 Multiple linear regression: accounting for multiple simultaneous determinants of a continuous dependent variable. V117; P1732).

# In my view, the authors' effort to show that the results of previous SSc studies were inadequate or inaccurate is more prominent than the purpose of reporting the results of their own study in an objective way. This turns into an effort to make the readers accept the accuracy of their own results rather than putting the originality of the article forward.

# Author’s response. It was not our intention to attempt to show that the results of previous studies were inadequate or inaccurate and we made no such claims in the manuscript. On the other hand, that there are numerous discrepancies between the findings about the left ventricle in the past SSc literature, as demonstrated in the tables, necessarily implies that some of the reported findings (negative or positive) are incorrect. This then was the rationale for our attempt to identify variables which may have confounded previous analyses. We wish to point out that this was our main study conclusion “In SSc there are associations of sex, body size, age and disease type with LV structural variables, of age and heart rate with E/A, and of age with both systolic and early diastolic LV long-axis velocities. Appropriate adjustment for these variables could help to resolve current uncertainties regarding SSc effects on the left ventricle.” We did not, and indeed could not, claim or prove the inaccuracy or inadequacy of any particular study. They findings do, however, provide a potential basis for performing appropriate adjustments in the analyses of future studies, and also in the reanalysis of previous data, with the possibilities that this could change previous false negative findings into true positives or previous false positive findings into true negatives.

We found it difficult to understand the reasoning behind the statement “This turns into an effort to make the readers accept the accuracy of their own results rather than putting the originality of the article forward”. We clearly believe in the accuracy of the findings of our study and that the results have been presented in an objective way. We are not aware of any other study which addresses the aims of the current study so we believe the originality of the study speaks for itself.

---

## [Editor Report · Decision Letter 2]

1 Oct 2021

Determinants of left ventricular structure, filling and long axis function in systemic sclerosis

PONE-D-21-07257R2

Dear Dr. Peverill,

We’re pleased to inform you that your manuscript has been judged scientifically suitable for publication and will be formally accepted for publication once it meets all outstanding technical requirements.

Kind regards,

Masataka Kuwana, MD, PhD

Academic Editor

PLOS ONE

Additional Editor Comments (optional):

This editor principally agree with the responses to the comments. The points raised by one of the reviewers are adequately reflected in the revised version.
---

## [Editor Report · Acceptance letter]

11 Oct 2021

PONE-D-21-07257R2 

Determinants of left ventricular structure, filling and long axis function in systemic sclerosis 

Dear Dr. Peverill:

I'm pleased to inform you that your manuscript has been deemed suitable for publication in PLOS ONE. Congratulations! Your manuscript is now with our production department. 

Kind regards, 

on behalf of

Prof. Masataka Kuwana 

Academic Editor

PLOS ONE